# In Silico Characterization and Analysis of Clinically Significant Variants of Lipase-H (LIPH Gene) Protein Associated with Hypotrichosis

**DOI:** 10.3390/ph16060803

**Published:** 2023-05-29

**Authors:** Hamza Ali Khan, Muhammad Umair Asif, Muhammad Khurram Ijaz, Metab Alharbi, Yasir Ali, Faisal Ahmad, Ramsha Azhar, Sajjad Ahmad, Muhammad Irfan, Maryana Javed, Noorulain Naseer, Abdul Aziz

**Affiliations:** 1Department of Computer Science and Bioinformatics, Khushal Khan Khattak University, Karak 27200, Pakistan; hamzaalikhan2211@gmail.com; 2Basic Health Unit, 196 RB, Faisalabad 38000, Pakistan; umair13047@gmail.com; 3Rural Health Center, Faislaabad 38000, Pakistan; khurramijaz205@gmail.com; 4Department of Pharmacology and Toxicology, College of Pharmacy, King Saud University, Riyadh 11451, Saudi Arabia; mesalharbi@ksu.edu.sa; 5National Center for Bioinformatics, Quaid-i-Azam University, Islamabad 45320, Pakistan; yasirkhanqu@gmail.com (Y.A.); faisalahmad@bs.qau.edu.pk (F.A.); ramshaazhar12@gmail.com (R.A.); maryanajaved@gmail.com (M.J.); noorulainnaseer423@gmail.com (N.N.); 6Department of Health and Biological Sciences, Abasyn University, Peshawar 25000, Pakistan; 7Department of Oral Biology, College of Dentistry, University of Florida, Gainesville, FL 32611, USA; irfanmuhammad@ufl.edu

**Keywords:** hair loss, Lipase-H, hypotrichosis, missense mutations, molecular dynamics simulation, alopecia

## Abstract

Hypotrichosis is an uncommon type of alopecia (hair loss) characterized by coarse scalp hair caused by the reduced or fully terminated activity of the Lipase-H (LIPH) enzyme. LIPH gene mutations contribute to the development of irregular or non-functional proteins. Because several cellular processes, including cell maturation and proliferation, are inhibited when this enzyme is inactive, the hair follicles become structurally unreliable, undeveloped, and immature. This results in brittle hair, as well as altered hair shaft development and structure. Because of these nsSNPs, the protein’s structure and/or function may be altered. Given the difficulty in discovering functional SNPs in genes associated with disease, it is possible to assess potential functional SNPs before conducting broader population investigations. As a result, in our in silico analysis, we separated potentially hazardous nsSNPs of the LIPH gene from benign representatives using a variety of sequencing and architecture-based bioinformatics approaches. Using seven prediction algorithms, 9 out of a total of 215 nsSNPs were shown to be the most likely to cause harm. In order to distinguish between potentially harmful and benign nsSNPs of the LIPH gene, in our in silico investigation, we employed a range of sequence- and architecture-based bioinformatics techniques. Three nsSNPs (W108R, C246S, and H248N) were chosen as potentially harmful. The present findings will likely be helpful in future large population-based studies, as well as in drug discovery, particularly in the creation of personalized medicine, since this study provides an initial thorough investigation of the functional nsSNPs of LIPH.

## 1. Introduction

Originating from the ectoderm of the skin, hair is one of the crucial components of the human body. It is a protective projection which, along with sebaceous organs or glands, nails, and sweat glands, is considered to improve skin structure [1]. Hair present on the dermis is a protein component that grows from follicles. Hypotrichosis is an uncommon type of hereditary alopecia that affects both males and females equally. This form of hair loss begins in early childhood. Hypotrichosis (hair loss) is caused by either LIPH or LPAR6 gene mutations [2]. A representative of the lipases gene family, the LIPH gene (OMIM 607365) is situated on the human chromosome at location 3q27.2. GRCh38.p12 and 92 annotations, comprising 10 exons, are included in Homo sapiens Annotation Release 109; this gene is approximately 46,396 bp long and is located at the molecular position 3: 185, 506, 262–185, 552, 588 on the reverse strand of chromosome 3.

This reverse gene transcribes from its mRNA 4001 bp cDNA; this produces a member of the Lipase H family, a 451 amino acid protein (LIPH En-zyme). To create 2-acyl lyso-phosphatidic acid (LPA) and fatty acids outside of a cell’s membrane, lipase H hydrolyzes phosphatidic acid (PA) [3]. Some phospholipids, such as phosphatidylserine (PS), phosphatidylcholine (PC), and phosphatidylethanolamine (PE) or triacylglycerol are not hydrolyzed (TG). So far, the coding sequence of the LIPH gene has been found to contain 295 distinct types of mutations, encompassing splice site mutations, deletions, and insertions. The LIPH gene-related disorder is woolly hair, or hypotrichosis (OMIM 604379). It is the hair condition that contributes to the growth of sparse hair in the scalp area of the head (hypotrichosis). The rate of incidence of this disease is rare in other areas of the body. Specific mutations in the LIPH gene are observed exclusively in particular racial communities, such as those with Japanese or Pakistani ancestry, or among the Mari and Chuvash populations in Russia, indicating the potential impact of founder effects [4,5].

LIPH gene mutations result in the development of an irregular protein or a protein with no function whatsoever. When this enzyme is non-functional, certain cellular functions, such as the maturation and proliferation of hair follicle cells, result in the non-formation of LPA. Such hair follicles are structurally irregular, underdeveloped, and immature, resulting in fragile hair and modified hair growth and structure. In individuals that have autosomal recessive hypotrichosis, the lack of Lipase-H enzymes may also often contribute to skin deformities. In this study, the nsSNPs (non-synonymous single nucleotide polymorphisms) of the LIPH amino acid chain were examined. In addition to this, the effect of the sequence variant on the LIPH protein has been recognized. Studies have also been conducted to examine the impact of the mutation on the role and durability of the LIPH amino acid chain.

The objective of the present study is to detect and predict deadly nsSNPs within the LIPH gene, along with their links to diseases and the ramifications of detrimental nsSNPs on the behavior and structure of the protein. Several bioinformatics tools, including SIFT [6], Polyphen2 [7], SNAP2, SNPs&GO [8], PhD-SNP [9], I-Mutant [10], and Mu-Pro [11], were employed to predict the pathogenic nsSNPs in the LIPH gene. Additionally, amino acid residue conservation was detected using ConSurf [12]. The structure was predicted using SWISS-MODEL and which was then further refined using ModRefiner [13]. To run molecular dynamics simulations and to observe structural changes over time, Schrodinger suite was employed. A thorough visualization study was carried out using PyMol [14].

## 2. Results

### 2.1. Retrieval of nsSNPs of the LIPH Gene

A variety of bioinformatics applications and datasets, such as the ensemble gene annotation tool and the dbSNP resource, were employed to identify effective SNPs, which can be situated at multiple locations on the gene, within the LIPH gene. This research examined the effects of nsSNPs on the structure and functionality of the Lipase-H amino acid chain encoded by the LIPH gene. Additional SNPs were found in other protein portions as well; while most of the identified SNPs were located within the intronic segment, our study centered on the exonic segment’s clinically significant SNPs, as depicted in Figure 1.

### 2.2. Screening of nsSNPs of LIPH Gene

The sorting intolerant from tolerant algorithm was utilized on 307 nsSNPs to ascertain the score for the tolerance index (TI) of the mutations. By aligning similar conserved amino acid sequences, SIFT determines the TI score, which ranges from 0 to 1. During ours initial examination, of 307 examples, SIFT detected 259 non synonymous SNPS, displaying TI scores between 0 and 0.3, with 199 nsSNPs exhibiting TI scores of 0, 22 showing TI scores of 0.1, 17 expressing TI scores of 0.2, and 21 revealing TI scores of 0.3, indicating the importance of these nsSNPs (Table 1). Out of the 259 isolated nsSNPs, 158 were designated by Polyphen2 as only “possibly damaging” or “probably damaging” SNPs. REVEL results revealed that 136 cases of illness were caused by the remaining 158 nsSNPs. MetaLR showed 128 as damaging, and finally, Mutation Assessor identified only two SNPs as damaging. These mutations are summarized in Appendix A.

### 2.3. Prediction of Clinically Significant Deleterious nsSNPs of LIPH Protein

The large dataset of predicted deleterious SNPs was narrowed down based on the clinical significance of all the mutations. The extensive literature studies suggested that only three nsSNPs (LIPHW108R, LIPHC246S, and LIPHH248N) be classified as clinically significant. These selected nsSNPs were analyzed using five resources—SNP&GO, Polyphen-2, PhD-SNP, SIFT, and PROVEAN—to check pathogenicity and to predict nsSNPs with a significant deleterious outcome affecting the structural character and the role of the LIPH amino acid chain. Each of the chosen nsSNPs were predicted to be adverse or pathogenic (Table 1).

### 2.4. Pathological Annotation of Clinically Significant Deleterious nsSNPs on LIPH Protein

The pathological implications of mutations on the LIPH protein were annotated using the PMUT service. The PMUT results reflected the PMUT scores of 0.93, 0.90, and 0.90 for mutations LIPHW108R, LIPHC246S, and LIPHH248N, respectively. These results show that there is around a 93% probability of LIPHW108R producing a pathological impact, and a 90% probability for LIPHC246S and LIPHH248N to convey an associated pathological impact. The prediction class was identified as “disease” for all three of the mutant sequences.

### 2.5. Influence of Medically Significant Variations on LIPH Protein Robustness

The protein structure stability was predicted through an online I-Mutant server (version 2.0). To express the outcome of I-Mutant 2.0, the reliability index (RI) and the free energy change value (DDG) were employed. The subjected three mutations yielded a stability decrease of (−1.91, 1.55, and −0.85) kcal/mol. The I-Mutant results confirmed that selected nsSNPs cause a decrease in the protein stability because they have a DDG value (<−0.5), which is a sign of a larger effect on the protein. These steep declines in the stability of the LIPH protein may cause the loss of LIPH function, which triggers hypotrichosis.

### 2.6. Evolutionary Conservation Analysis of LIPH Protein

To determine the evolutionary conservation of the modified protein configurations, the ConSurf computational server was employed. Out of the three nsSNPs, two amino acids (LIPHW108R, LIPHC246S) were detected in the LIPH gene as functional, highly conserved, and exposed, while one nsSNP amino acid (LIPHH248N) was shown to be strongly conserved and prominently displayed (Table 2). LIPHW108R and LIPHC246S were considered to play a structural role, while LIPHH248N was predicted to perform a functional role. The conservation score was predicted as 9 for all the variants (Figure 2).

### 2.7. Comparative Analysis of LIPHWT and LIPHMT Structures

To determine whether the high-risk nsSNPs selected had a significant impact on the final proteins, predicted 3D modeling and comparisons of structure between mutant models and wild-type were used. SWISS-MODEL was used to predict the model for LIPH protein. SWISS-MODEL provided 53 templates, but we selected 2ppl.1, which is the highest query coverage value, and PyMol was used to construct the LIPHMT models. The predicted model, along with the mutation position, was visualized by PyMol. The results were then confirmed by the nsSNPs (Figure 3). To check the structural comparison of LIPH protein with their mutant model, TM-align server was used. For every mutant model with a TM-score of 1 and an RMSD of 0, the TM-align server reported no major deviations from their unmutated structures (Table 3). The generated mutant model was validated through ERRAT (Table 3). The structural integrity was further validated by the Ramachandran map derived from calculated dihedral angles. The PDB inputs, both mutant and non-mutant, were determined by PROCHECK. The region with the highest frequency of occurrence in LIPHWT contains 340 residues (80.3%), while the additionally allowed region contains 39 residues (9.9%), the generously allowed region contains 7 residues (1.8%), and the disallowed region contains 8 residues (2.0%). One of the most detrimental mutations, LIPHW108R, includes 350 residues (88.8%) in the region that is most preferred, 29 residues (7.4%) in the region that is additionally allowed, 7 residues (1.8%) in the region that is liberally allowed, and 8 (2.0%) elsewhere. LIPHC246S exhibits 341 residues (86.5%) in the region with the highest preference, 40 residues (10.2%) in the region with additional permission, 8 residues (2.0%) in the region with liberal permission, and 5 residues (1.3%) in the region with the lowest preference. As evidence that the mutant models were accurately modeled, LIPHH248N exhibits 342 residues (86.8%). In the most favored region, approximately 70% of the residues were found to be favorable, 10.2% of the residues were in the additionally allowed zone, 1.8% of the residues were in the generously allowed region, and 1.3% of the residues were in the banned regions (Table 3).

### 2.8. Stability Analysis of LIPHWT and LIPHMTs Protein Models

The molecular dynamics simulation analysis was conducted to assess the **RMSD**, **RMSF**, and the percentage of amino acid secondary structures. Figure 4A presents the changes in RMSD values over time for the backbone atoms of the non-mutant and mutant proteins. The plots displaying the RMSD of both proteins determine that the protein structure attains equilibrium at 10 ns. During the simulation timeframe, the RMSD value for the predicted protein fluctuated up to 20 ns, with an average value of 1.2 Å. Subsequently, the RMSD values remained within 1.2 Å, which is highly suitable for small predicted proteins, as demonstrated in Figure 4A. For mutant proteins, the RMSD is higher, which indicates major structural and functional changes.

### 2.9. Flexibility Analysis of LIPHWT and LIPHMTs

The flexibility analysis of each residue was measured by root mean square fluctuation (RMSF) analysis. The average RMSF value for LIPHWT was observed to be 0.24 Å, whereas the average RMSF values for mutant LIPHW108R, LIPHH248D, and LIPHC246S were (0.24, 0.27, and 0.22) Å, respectively. The norm RSMF range of LIPHWT and all LIPHMTs revealed that all were stable during the simulation period of 200 ns. The residue Pro234 exhibited higher fluctuations in LIPHW108R and LIPHH248D, whereas in LIPHWT, the residue Phe25 showed the highest fluctuation. Furthermore, the residue His305 was the most fluctuating residue in LIPHC246S, as shown in Figure 4B.

### 2.10. Gyration Analysis of LIPHWT and LIPHMTs

The conformational behavior of the protein was evaluated by analyzing the system’s radius of gyration values. A lower Rg value is indicative of a more compact amino acid architecture. The norm Rg value for LIPHWT was 27.10 Å, whereas for mutant LIPHW108R, LIPHH248D, and LIPHC246S, it was 27.08 Å, 26.92 Å, and 26.35 Å, respectively. The lowest Rg value was obtained for LIPHC246S, which reveals that LIPHC246S showed the most compactness (Figure 4C).

### 2.11. Hydrogen Bond Occupancy

In this study, hydrogen bond profiles of the proteins were also examined, as shown in Figure 4D. From the plot in Figure 4D, it is observed that the proteins exhibited similar hydrogen bonding. The hydrogen chemical link plays a complicated role in the durability of the amino acid. The range of hydrogen bonding of all proteins was observed between 259–361. The LIPHC246S exhibited the denser hydrogen bonding, i.e., ranging from 264–358, whereas for LIPHWT, LIPHW108R, and LIPHH248D, the bonding lay within the range of 269–353, 268–361, and 259–348, respectively.

### 2.12. Solvent-Accessible Surface Area of LIPHWT and LIPHMTs

The aqueous accessibility superficial area measures the portion of a bimolecular surface that is available to nearby aqueous molecules. The LIPHWT and LIPHMTs underwent SASA analysis. The mean SASA value of LIPHWT and the mutants LIPHW108R, LIPHH248D, and LIPHC246S were recorded as 2189.06 Å^2^, 2142.5 Å^2^, 2228.37 Å^2^, and 2097.09 Å^2^, respectively, throughout the simulation (Figure 4E). During the initial stages of the simulation, all proteins demonstrated a higher **SASA** value, but over time, it decreased, as seen in Figure 4E. The mutant LIPHH248D showed the highest mean value of SASA, i.e., 2228.37 Å^2^. The SASA calculations illustrated that mutant protein LIPHC246S showed decreased SASA value, thus further indicating that the mutant protein LIPHC246S may not have maintained contact with the solvent molecules around it.

### 2.13. Conformational Analysis of LIPHWT and LIPHMTs

Understanding proteins at the molecular level requires a comprehension of their structure and dynamics. Most of the conformational changes occurred at the loops at the terminals (Figure 5). The conformational analysis of LIPHWT at different time scales of simulations revealed certain structural changes. The helix at Ser14–Ser16, which was observed at the beginning of the simulation, disappeared at 80 ns and never again formed during the simulation. The conversion of many loops (Arg42–Tyr46, Asn102–Asp107, Met313–Asp319, Ser361–Thr368, Phe441–Leu445, and Glu448–Gln450) to beta sheets occurred at different sites in order to attain stability. All these beta sheets formed at 40 ns and remained until the end of the simulations, except for Glu448–Gln450, which again converted to a loop at 120 ns. In LIPHW108R protein, the conformational changes were observed in the loop regions, as loops are the most flexible structures. In LIPHW108R protein, certain loops are converted into helices (Leu401–Arg405 and Asn383–Leu386). In all LIPHMTs, the formation of beta sheets at Gln55–Ile57 and Val440–Ile444 was observed during simulation, whereas in LIPHH248N, the beta sheet Val440–Ile444 again converted into the loop at the end of the simulation. In LIPHW108R, the protein beta sheet (Gln425–Cys427) and the helix (Tyr297–Asn301) converted into loops. The conversion of the helix (His184–Asp191 and Pro195–Asp197) to a loop occurred in both LIPHC246S and LIPHH248N. In LIPHC246S, the formation of a new helix at Ile237–Gly239 was observed (Figure 5).

#### Comparative Functional Analysis of LIPHWT and Mutant Models

Functional displacement of every system was calculated using the DCCM method as a function of time. The results show a substantial negative connection between two clusters of LIPH residues (Figure 6A). The LIPHWT (cluster 2; 200aa–320aa) and the mutants LIPHW108R and LIPHC246S showed a relatively comparable correlation, demonstrating that the variation in correlation may be caused by a shift in how these two proteins are confirmed following a point mutation. Moreover, all of the mutants correlated differently from the LIPHWT in cluster 1 (as seen in Figure 6A). In contrast to LIPHWT, other mutants, such as LIPHH248N, showed an entirely different association among their local moments (Figure 6D). In general, the DCCM findings demonstrate that both original amino acid and the mutant amino acid manifest distinctive arrangements of strong agreement and disagreement. While the red hue represents a significantly favorable connection, the dark blue hue indicates a strong negative link to a few of the leftovers. Residues that are positively associated travel in the same direction, but those that are negatively correlated move in opposite directions.

### 2.14. Dimensionality Reduction Using PCA

An investigation of the dynamically advantageous structural alterations in the chemistry of LIPHWT and its four variants (LIPHW108R, LIPHC246S, and LIPHH248N) was conducted using motion mode analysis. The variables for the **PCA** involved the coordinate covariance matrix (CCM), which was calculated from the time-dependent data of 3D positional coordinates of multiple variant models during the 200 ns MD simulation. This analysis is commonly known as PCA. The results show that the LIPHWT system and its four variations (LIPHW108R, LIPHC246S, and LIPHH248N) all exhibited varied arrangements and failed to converge into a single energy level, demonstrating that the mutant forms of the LIPH amino acid had an unstable arrangement of configurations (Figure 7).

## 3. Discussion

Hair is included in those important parts of the human body originating from the ectoderm of human skin. Hair is a protective projection on the human body, and it is considered to be skin accessory, along with sebaceous organs or glands, nails, and sweat glands [1]. Hair presenting in the dermis is a protein component that grows from follicles. LIPH gene mutations result in the development of an irregular protein. When this Lipase H is non-functional, certain cellular functions, such as the maturation and proliferation of hair follicle cells, results in the non-formation of LPA. Such hair follicles are structurally irregular, underdeveloped, and immature, resulting in fragile hair and altered hair growth and structure. In individuals that have autosomal recessive hypotrichosis, the lack of Lipase-H enzymes may also often contribute to skin deformities.

Non-synonymous SNPs are created when an amino acid is mutated in a protein chain, impacting the protein’s composition and purpose. Many genetic abnormalities are caused by SNPs. The functionally relevant area is at significant risk of mutation due to amino acid degeneracy. Finding the amino acid that significantly influences disease development and differentiating between beneficial and harmful SNPs are both very challenging tasks [15]. When an amino acid in a protein chain changes, a non-synonymous single nucleotide polymorphism (SNP) occurs that affects the protein’s structure and function. SNPs can be linked to common genetic disorders [16,17]. Mutations are common in the functionally important region due to amino acid degeneracy. Distinguishing between beneficial and deleterious SNPs is difficult, as is determining the amino acid that has a significant impact on disease predisposition.

Bioinformatics has been impactful in the field of genome-wide association studies (GWAS) for mutation detection [18,19], annotation [18,20],, and impact [20,21,22,23], as well as the discovery of new and effective medications [24,25]. Important in silico approaches include those used to identify the location genetic locus, forecast its transcripts, and predict its interactions with other genes and proteins [26], as well as those used to evaluate the protein’s cellular function and structure [27]. We can distinguish between benign and harmful SNPs using in silico research [27], which employs a variety of algorithms and publicly available databases. An examination of the altered amino acids using structural and phylogenetic data produced very precise results. NSSNPs in the amino acid’s coding plane can cause amino acid substitutions that alter the influence of amino acids on biological functions and heighten disease susceptibility [28]. The detection of harmful nsSNPs through classifying tolerant/intolerant forms is unrivaled when it comes to researching an individual’s susceptibility to disease. However, not every mutation has a negative effect on function, and some may be well tolerated. The molecular signature of nsSNPs linked to a certain disease and other traits has previously been demonstrated. As a result, these nsSNPs may interfere with protein–protein interactions, affect enzyme activity, and ultimately lead to a change in protein structure. The identification and characterization of functionally connected nsSNPs, as well as their separation from non-functionally related nsSNPs, is the current focus of molecular biology [29,30,31] and population genetics.

In the present research, we used in silico methods to perform various analyses of the nsSNPs, since a mutation in the LIPH gene is accountable for autosomal recessive hypotrichosis. Alopecia is a rare sub-type of hair loss that causes hair to become scarce (hypotrichosis) on the scalp skin. At position 27.22, The LIPH gene or amino acid is situated on chromosome 3 (eq27.2). The typical function of lipase is to hydrolyze PA outside the cell membrane to produce 2-acyl lysophosphatidic acid (LPA, a potent bioactive lipid mediator) and fatty acid. In this gene, two mutations (**c.2T > C; p.M1T and c.322T > C; p. W108R**) have been identified in both families, causing autosomal recessive hypotrichosis. LIPH gene mutations result in the development of an irregular protein or a protein with no function. When this enzyme is non-functional, certain cellular functions, such as the maturation and proliferation of hair follicle cells, result in the non-formation of LPA. Such hair follicles are structurally irregular, underdeveloped, and immature, resulting in fragile hair and altered hair shaft growth and structure. These hairs are weak and quickly split. In individuals who have autosomal recessive hypotrichosis, a lack of Lipase-H enzymes can also often contribute to skin deformities (NIH). In this research, an attempt is designed to identify SNPs that modify the architecture and operation of the coded Lipase-H protein. The gene LIPH encodes the enzyme. Using various bioinformatics computational protein prediction methods, we tested the pathogenicity of mutation (c.322T > C; p. W108R). SIFT, POLYPHEN-2, PROVEAN, SNP&GO, I-MUTANT, and PhD-SNP were used to describe the SNP causing the illness by classifying the mutations. Both (**c.736T > A, p. C246S**) and (**c.744 C > A; p. H248N**) are deleterious and likely to be toxic, according to the evaluations using SIFT and the Polyphen2 tool [32]. Using I-Mutant version 2.0 to verify the durability of the mutant protein, it was shown that the amino acid has a lack of durability due to the disease that is induced. Structural validation and comparison are achieved with the assistance of molecular dynamic simulation, in which a strong distinction between the protein of the mutant and wild form has been revealed.

## 4. Materials and Methods

### 4.1. Retrieval of LIPH Sequence and nsSNPs

The primary sequence of the LIPH protein, with a length of 451aa in the FASTA format, was retrieved from browsing the ensemble genome website (https://asia.ensembl.org/index.html) (accessed on 3 February 2023). We retrieved all the nsSNPs found in the dbSNP and Ensembl genome browser.

### 4.2. Identification of Deleterious nsSNPs

The predictions of damaging nsSNPs affecting the function of LIPH protein were developed using different bioinformatics equipment: **SIFT** (sorting intolerant from tolerant—http://sift.jcvi.org) (accessed on 16 March 2023) [33], **PROVEAN** (protein or amino acid variation effect analyzer) [34], **Polyphen-2** (polymorphism phenotypingv2) [35], and **SNP&GO** (single nucleotide polymorphism and gene ontology), which are linked using **PhD-SNP** (predictor of human deleterious single nucleotide polymorphism—https://snps.biofold.org/snps-and-go/snps-and-go.html (accessed on 16 March 2023)) [36]. The non-synonymous single nucleotide polymorphisms (nsSNPs) that were predicted to be harmful by all five in silico tools were identified as “high-risk” nsSNPs and were selected for further examination.

### 4.3. Validation of Deleterious nsSNPs of LIPH Gene

PMut served as a means to confirm the pathogenicity of the identified suspected harmful nsSNPs (http://mmb.irbbarcelona.org/PMut/ (accessed on 16 March 2023)) [37]. For 12,141 proteins, this artificial neural network-based approach contains 27,203 dangerous and 38,078 benign mutations. The prediction percentage was calculated, along with a score that predicts the likelihood of an event ranging from 0 to 1. The **nsSNPs** of a score with values of 0.5 or less may be considered neutral, whereas those score with values of >0.5 are considered disease associated [38].

### 4.4. Structure Stability of Mutant Protein

I-Mutant V2.0 was employed to forecast the protein stability of the nsSNPs (https://folding.biofold.org/i-mutant/i-mutant2.0.html (accessed on 16 March 2023)). This method evaluates the corresponding values of free energy transform when determining the increase or decrease in durability transformation in the mutant proteins (DDG). I-Mutant version 2.0 employs an SVM machine learning algorithm and ProTherm-derived information, or a dataset, which is a more detailed collection of databanks that contain experimental thermodynamic data on mutant protein stability. This web server additionally generated a reliability index (RI) ranging from the lowest reliability (0) to the highest reliability (10), in addition to the other forecasts [10].

### 4.5. Structure Prediction and Validation

The determination of a 3D protein model by NMR or X-ray crystallography methods is very expensive; thus, the homology modeling technique, which is less time consuming and highly accurate, can be used for the prediction of a 3D protein model. For systematic comparative modeling of 3D amino acid structures based on models and accessible through an offline script-based tool, Modeler was used [39]. PyMOL is a free desktop-based software. It is used for protein visualization. Using various basic functional parameters, it manipulates the structure to evaluate the chemical properties or proteins. In general, the tool is available for operating systems such as Linux, Windows, and macOS [40]. ERRAT is an online bioinformatics tool used for the validation and refinement of protein 3-D structures. After prediction, the quality of the protein structure was analyzed using the ERRAT tool [41]. To further check the estimate of the structure quality of the proteins, PROCHECK Ramachandran plots were used (https://saves.mbi.ucla.edu/ (accessed on 16 March 2023)).

### 4.6. Computer Simulation of Molecular Dynamics

MD simulations were carried out for 200 ns using Desmond, a Schrödinger LLC platform. The initial structures of the proteins were obtained from the predicted models. The amino acid architecture was preprocessed using Maestro’s Protein Preparation Wizard, which revealed protein refinement, as well as reduction. The Device Builder tool was used to prepare all structures. The orthorhombic box solvent model was employed, using TIP3PP. In the simulation, the OPLS-2005 force field was used [42]. The system was neutralized by adding counter ions, and 0.15 M salt (NaCl) was applied to mimic physiological conditions. The simulation was carried out under the NPT ensemble at a temperature of 300 K and a pressure of 1 atm. After every 50 ps, the trajectories were saved for further assessment, and the robustness of the simulations was assessed by measuring protein RMSD, RMSF, and the radius of gyration.

### 4.7. Dynamic Cross-Correlation Map (DCCM)

A DCCM plot was created to investigate the cross-correlation shift of the backbone atom (C) and to offer insight into how mutations affect the dynamics of LIPHWT and LIPHMTs. DCCM was created to analyze and determine the dominating correlated movements of each residue in distinct LIPH systems.
C_ij_ = 〈∆ri. ∆rj〉
(〈∆ri〉2 < ∆rj < ∆ri > 2) ½
Here, the matrix C_ij_ illustrated the time-correlated information between the (i) and (j) atomic particles of a protein, where Δri and Δrj were the displacements from the mean position of the ith and jth residues with respect to time. The angular brackets “〈〉” represented the time norm over the entire route. The Cij ranged from −1 to 1. The only alpha carbon atoms from the last 5000 snapshots were selected at 0.002 ns time intervals to create the matrix. The following displayed graph from DCCM is based upon two positive (+ve) and negative (−ve) values; the (+ve)-values explain the residue motion in a unique direction, while the (−ve) values indicate the residue displacement in the opposite direction.

### 4.8. Essential Dynamics Using Principal Components Analysis

Using the trajectories files produced by MD simulation, we examined the dynamic movements of every system, including bound GalN6SWT, bound GalN6SMTs, free GalN6SWT, and free GalN6SMTs, using principal component analysis (PCA) or conformational sampling [24]. The rotational and translational motions of the positioned coordinates were removed before the system was placed on a reference structure. Subsequently, the positional covariance matrix (PCM) of the atomic coordinates and their eigenmode/eigenvector was constructed. As a result, the retrieved data for the matrix was diagonalized by the orthogonal coordinate transformation matrix, yielding the diagonal matrix of the eigenvalues. The first eigenvector and its associated eigenvalue represent the protein structure ensemble’s significant dominating motion. Matplotlib was used to display the dominant motion of all systems.

## 5. Conclusions

In this study, the LIPH gene was examined using a computational technique to determine the significance of harmful SNPs in the coding and untranslated zones. Based on clinical importance, 3 out of a total of 307 missense SNPs were projected to be the most harmful. These SNPs were anticipated to be highly conserved and to influence protein stability. The structural analysis found that all W108R, C346S, and H248N mutations exhibited significant RMSD values, indicating a loss of total hydrogen bonds. As a result, we determined that the three nsSNPs W108R, C346S, and H248N, might be key players in the development of hypotrichosis, a rare type of alopecia.

## Figures and Tables

**Figure 1 pharmaceuticals-16-00803-f001:**
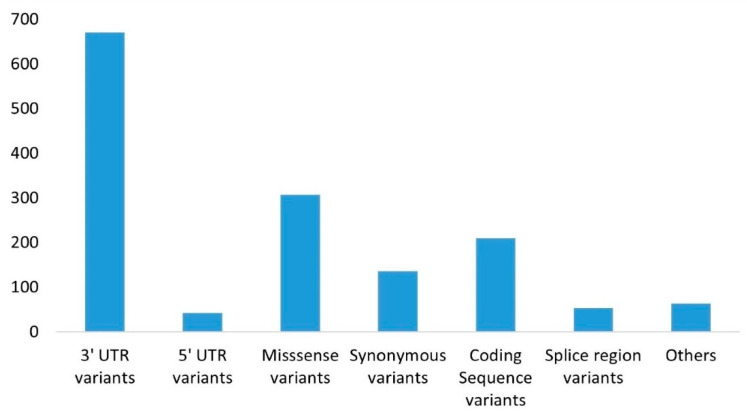
The distribution of different variants of the LIPH gene is based on their locus and type of mutation: 3′ UTR exhibited the most variations, while 5′ UTR underwent the least number of mutations.

**Figure 2 pharmaceuticals-16-00803-f002:**
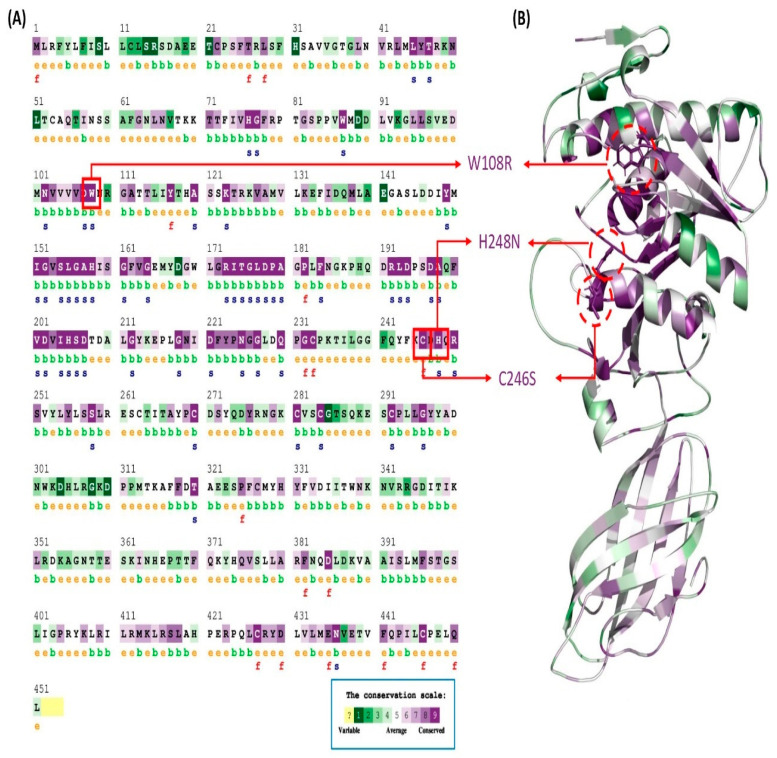
ConSurf sequence conservation prediction of the LIPH amino acid. (**A**) The sequence conservation analysis of the LIPH protein with color-coded conservation and residues; structural and functional impact labeled with “s” and “f”, respectively. (**B**) The conserved residues are depicted via a color-coded cartoon structure representation.

**Figure 3 pharmaceuticals-16-00803-f003:**
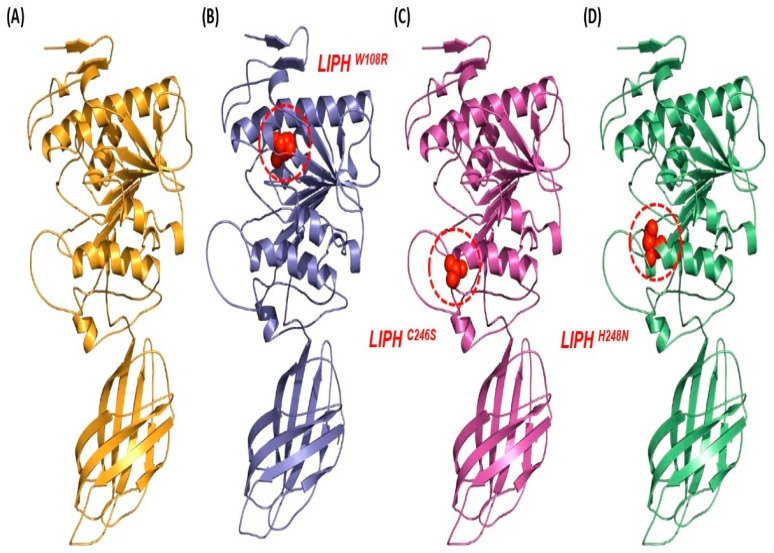
Mutation mapping and comparative analysis of LIPHWT with LIPHMT structures created using PyMol v2.5. (**A**) A 3D structure of LIPHWT protein. (**B**) Tryptophan to arginine substitution mapped at 108th position. (**C**) Cysteine to serine mutation mapped at 246th position. (**D**) Histidine to asparagine substitution mapped at 248th position.

**Figure 4 pharmaceuticals-16-00803-f004:**
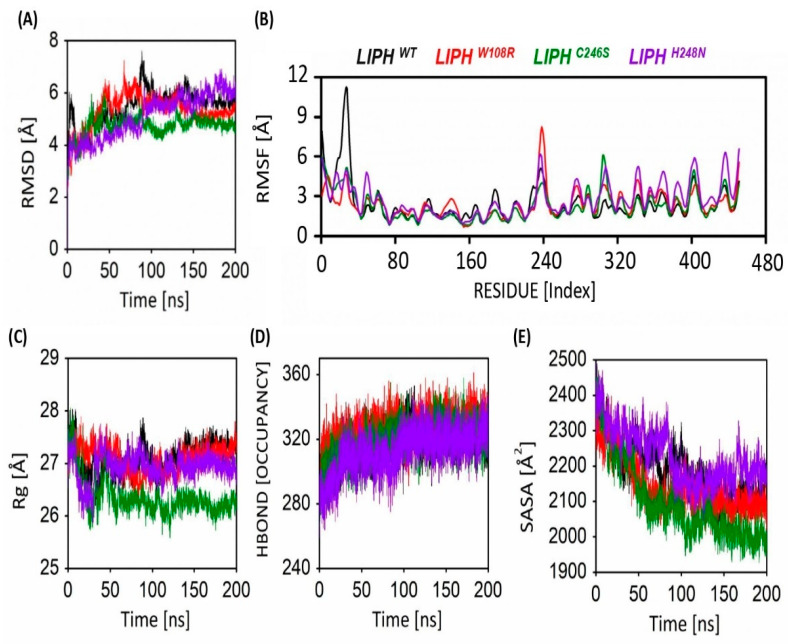
Post MD analysis of carbon alpha (Ca) atoms of LIPHWT and LIPHMTs protein models. (**A**) RMSD, (**B**) per-residue RMSF, (**C**) gyration analysis, (**D**) hydrogen bond occupancy, and (**E**) SASA of the Ca atoms of LIPHWT and LIPHMTs protein models shown via color-coded line graphs for a 200 ns simulation time.

**Figure 5 pharmaceuticals-16-00803-f005:**
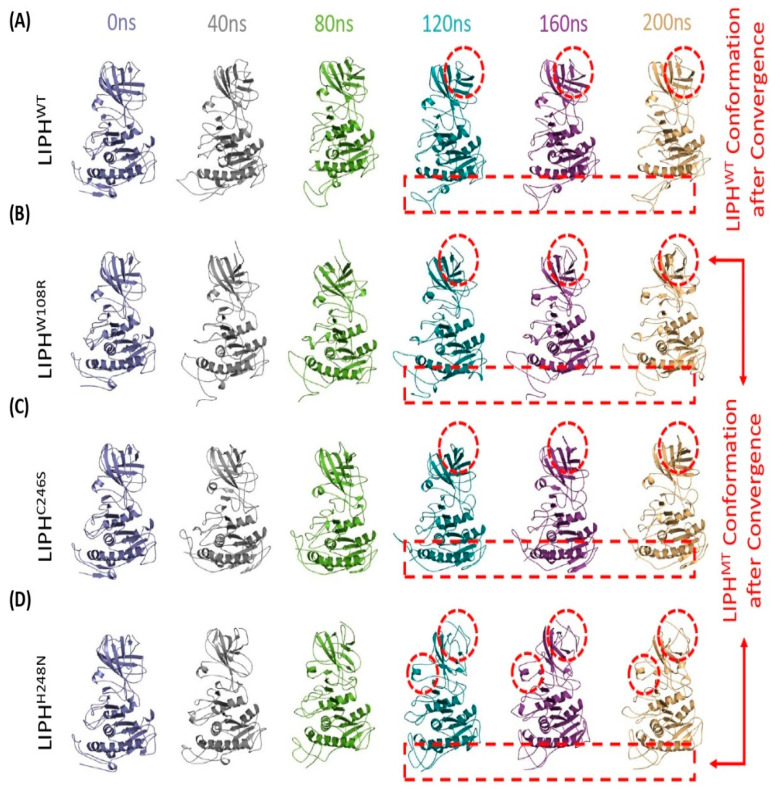
Conformational analysis of (**A**) LIPHWT, (**B**) LIPHW108R, (**C**) LIPHC246S, and (**D**) LIPHH248N protein models via color-coded protein structures obtained from the 200 ns trajectory at 0 ns, 40 ns, 80 ns, 120 ns, 160 ns, and 200 ns, respectively. The protein models were visualized using the PyMol molecular graphics system, version 2.5.

**Figure 6 pharmaceuticals-16-00803-f006:**
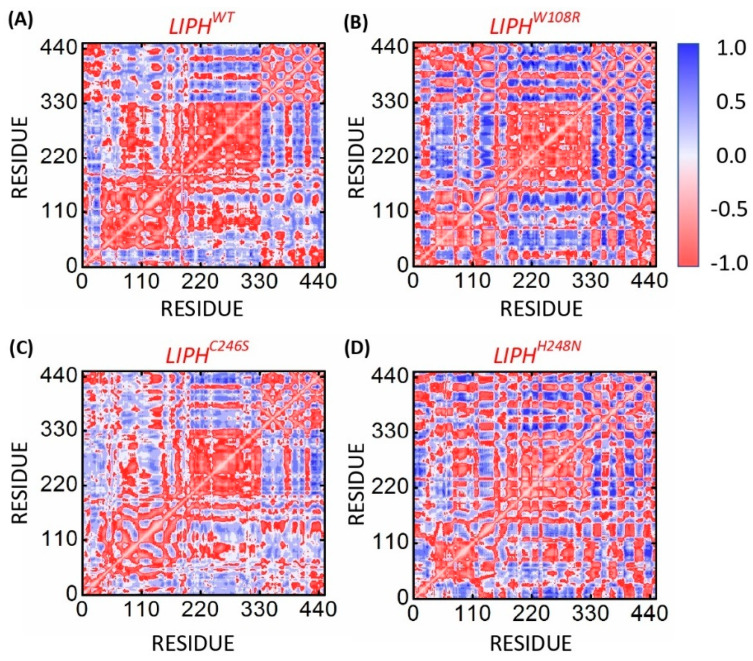
LIPHWT, LIPHW108R, LIPHC246S, and LIPHH248N dynamic cross-correlation matrices were created from 200 ns of MD trajectories. The blue hue represents the positive correlation of the local moments, while a negative association is represented by the color red. The (**A**–**D**) represent LIPH^WT^, LIPH^W108R^, LIPH^C246S^ and LIPH^H248N^ systems, respectively.

**Figure 7 pharmaceuticals-16-00803-f007:**
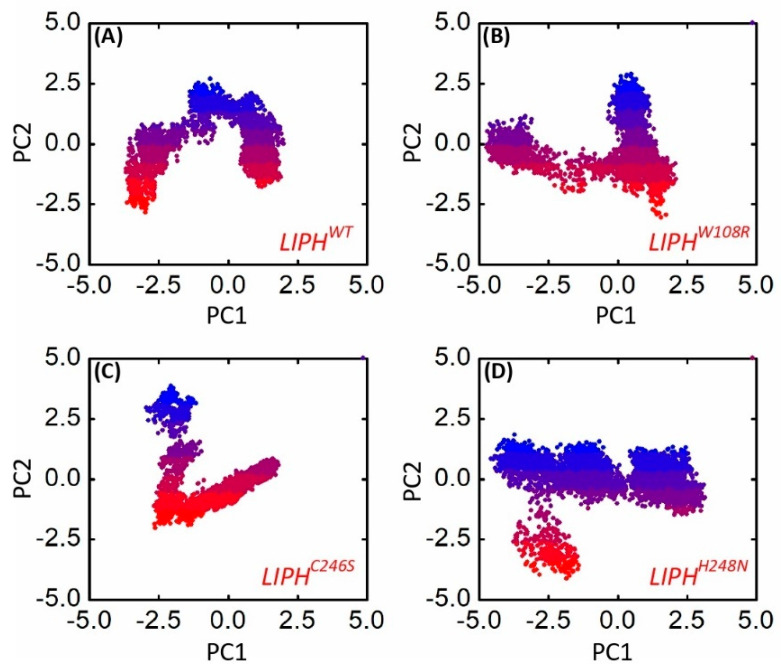
The conformational projection of LIPH and its mutations. (**A**) LIPH^WT^, (**B**) LIPH^W108R^, (**C**) LIPH^C246S^, and (**D**) LIPH^H248N^ mutant principal component analyses (PCAs) after 200 ns of MD trajectories. The blue to red color demonstrates frames picked from start to end of simulation time.

**Table 1 pharmaceuticals-16-00803-t001:** The deleterious nsSNPs were discovered in the LIPH gene using different tools.

Variant ID	AA Change	SIFT		Polyphen-2		PROVEAN		SNP&GO		PhD-SNP	
		Pred ^1^	Scr	Effect ^2^	Scr	Pred ^3^	Scr	Pred ^4^	RI	Pred ^5^	RI
rs267607219	W108R	Dmg	0.00	Pro. Dmg	1.00	Del	−12.81	Disease	8	Disease	9
rs201249971	C246S	Dmg	0.00	Pro. Dmg	1.00	Del	−9.36	Disease	7	Disease	9
rs201868115	H248N	Dmg	0.00	Pro. Dmg	1.00	Del	−6.56	Disease	7	Disease	9

AA (amino acid); Pred (prediction); Dmg (damage); Pro. Dmg (probably damage); Scr (score), and RI (reliability index). ^1^ SIFT: Dmg (Scr ≤ 0.05), ^2^ Polyphen-2: Pro. Dmg (Scr = 1.00), ^3^ PROVEAN: Del (Scr < −6.56), PhD-SNP: Disease (Scr > 0.5), ^4^ SNP&GO: Disease (Probability > 0.5) and ^5^ PhD-SNP: Disease (probability 0–9).

**Table 2 pharmaceuticals-16-00803-t002:** I-Mutant version 2.0, PMut, and ConSurf predictions of high-risk nsSNPs in LIPH protein.

Variant ID	AA Change	PMut		I-Mutant 2.0		ConSurf	
		Score and Percentage	Pred	Stability	DDG Value	Cons. Score	Pred
rs267607219	W108R	(0.93) 94%	Disease	Decrease	−1.91	9	Strongly conserved and prominently displayed (s)
rs201249971	C246S	(0.90) 93%	Disease	Decrease	−1.55	9	Strongly conserved and prominently displayed (s)
rs201868115	H248N	(0.90) 93%	Disease	Decrease	−0.85	9	Strongly conserved and prominently displayed (f)

RI (reliability index); DDG (value of free energy change); f (function amino acid, highly conserved and exposed), and s (structure amino acid, highly conserved and unexposed). PMut: Disease (Scr > 0.05), ConSurf: highly variable (Scr = 1.00), and highly conserved (Scr = 9).

**Table 3 pharmaceuticals-16-00803-t003:** Analysis of selected model: TM score, RMSD, ERRAT, and PROCHECK.

Model	TM-Align ^1^		ERRAT ^2^	PROCHECK Ramachandran Plot Analysis ^3^			
	TM Score	RMSD	ERRAT Value (Overall Quality Factor)	Residues in Most Favored Regions	Residues in Additional Allowed Regions	Residues in Generously Allowed Regions	Residues in Disallowed Regions
LIPH^WT^	Nill	Nill	90.26%	340 (80.3%)	39 (9.9%)	7 (1.8%)	8 (2.0%)
LIPH^W108R^	1.00	0.00	85.00%	350 (88.8%)	29 (7.4%)	7 (1.8%)	8 (2.0%)
LIPH^C246S^	1.00	0.00	85.00%	341 (86.5%)	40 (10.2%)	8 (2.0%)	5 (1.3%)
LIPH^H248N^	1.00	0.00	87.41%	342 (86.8%)	40 (10.2%)	7 (1.8%)	5 (1.3%)

Herein, “^1^” was used for structure similarities (**0.0 < TM score < 0.30**) and unordered configurations similarities (**0.50 < TM score <1.00**); the structure will belong to same fold. “^2^” ERRAT, which is the best reliable model, was used to evaluate the reliability of the model (ERRAT score > 85%). “^3^” PROCHECK was used to check the number of residues (percentage of residue) in the most favored regions, additionally allowed region, generously allowed region, and disallowed region.

## Data Availability

The text and its supporting information files contain all appropriate data.

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
