# Peer review of "In Silico Characterization and Analysis of Clinically Significant Variants of Lipase-H (LIPH Gene) Protein Associated with Hypotrichosis"

_pharmaceuticals, 2023, doi:10.3390/ph16060803_

Round 1

Reviewer 1 Report

Dear authors,

After the review process, I have several comments: the paper should be in the journal format (eg, section 4.3); you should define controls; the statistical details should be clearly inserted in section 4.

Best regards!

Author Response

Response to Reviewer Comments

We thanks the Referees for spending their time and interest in our work. We have checked all the comments and have made necessary changes accordingly.

Reviewer 1

After the review process, I have several comments: the paper should be in the journal format (eg, section 4.3); you should define controls; the statistical details should be clearly inserted in section 4.

Response

Thanks for the comments. Yes, we have made changes to the overall manuscript and have been organized accordingly with journal format. The control has been defined in the 4.3 section. Statistical details are not necessary as the whole study is based on computational approaches where we have already defined the mode of action and the confirmational changes throughout the simulation time intervals.

Reviewer 2 Report

In this study, the authors used bioinformatics approaches to study variants of Lipase-H (LIPH gene) protein associated with Hypotrichosis. The authors retrieved the primary sequence of LIPH protein fand SNPs rom Ensemble Genome browse, then using in-silico analysis, to separate potentially hazardous nsSNPs of the LIPH gene from benign ones using a variety of sequencing and architecture-based bioinformatics approaches. The authors chose 3 nsSNPs (W108R, C246S, and H248N) were chosen as potentially harmful. 

Major points

1) The quality of figures are poor, the authors need to provide high resolution ones.

2) This study is a predicative one. The authors need to confirm these findings on patients samples.

3) How can these 3 SNPs affect the therapy against Hypotrichosis.

4) How can these 3 SNPs affect LIPH function?

5) Did these 3 SNPs are found in any hypotrichosis cohort previously published? what are the percentages? please verify this part.

Moderate language editing

Author Response

Response to Reviewer Comments

We thanks the Referees for spending their time and interest in our work. We have checked all the comments and have made necessary changes accordingly.

Reviewer 2

Comments and Suggestions for Authors

In this study, the authors used bioinformatics approaches to study variants of Lipase-H (LIPH gene) protein associated with Hypotrichosis. The authors retrieved the primary sequence of LIPH protein fand SNPs rom Ensemble Genome browse, then using in-silico analysis, to separate potentially hazardous nsSNPs of the LIPH gene from benign ones using a variety of sequencing and architecture-based bioinformatics approaches. The authors chose 3 nsSNPs (W108R, C246S, and H248N) were chosen as potentially harmful. 

Major points

  • The quality of figures are poor, the authors need to provide high resolution ones.

Response

Thank you for highlighting. We have improved the quality and resolution of all the figures in the revised manuscript.

  • This study is a predicative one. The authors need to confirm these findings on patients’ samples.

Response

Thank you for the response. Yes, you are right, the whole study is based on computational approaches by predicting mode of action among wild type and mutated type Lipase-H (LIPH gene). We are currently working to find out the outcomes in the patients’ samples, but it will be incorporated in another manuscript which is based on experimental approaches.

  • How can these 3 SNPs affect the therapy against Hypotrichosis.

Response

Thank you for the comment. We have addressed the comment properly in the discussion section of line 385-389 at page 12 of the revised manuscript.

  • How can these 3 SNPs affect LIPH function?

Response

 Thank you for highlighting. The mentioned comment has been addressed accordingly in the discussion section of line 390-392 in the revised manuscript.

  • Did these 3 SNPs are found in any hypotrichosis cohort previously published? what are the percentages? please verify this part.

Response

Yes, the 3 SNPs are found in any hypotrichosis cohort previously published, but the percentage has been given based on patient’s sample in each study. Thus, it is not possible for us to link the different percentages into one frame in this study.

Reviewer 3 Report

It is not completed manuscript. For example, references are not listed although there are many citations. This manuscript is not at stage for external review. The editor has to be well checked the manuscript before sending it to external reviewers.

Author Response

Response to Reviewer Comments

We thanks the Referees for spending their time and interest in our work. We have checked all the comments and have made necessary changes accordingly.

Reviewer 3

Comments and Suggestions for Authors

It is not a completed manuscript. For example, references are not listed although there are many citations. This manuscript is not at the stage for external review. The editor has to be well checked the manuscript before sending it to external reviewers.

Response

Thank you for highlighting this. Yes, we have revisited the whole manuscript and have changed all the citations to a proper format. Corrections have been made in the revised manuscript.

Round 2

Reviewer 3 Report

Replies and revisions are fine. The revised version becomes acceptable.